# Cloth-aware Augmentation for Cloth-generalized Person Re-identification

## ABSTRACT

Person re-identification (ReID) is crucial in video surveillance, aiming to match individuals across different camera views while cloth-changing person re-identification (CC-ReID) focuses on pedestrians changing attire. Many existing CC-ReID methods overlook generalization, crucial for universality across cloth-consistent and cloth-changing scenarios. This paper pioneers exploring the cloth-generalized person re-identification (CG-ReID) task and introduces the Cloth-aware Augmentation (CaAug) strategy. Comprising domain augmentation and feature augmentation, CaAug aims to learn identity-relevant features adaptable to both scenarios. Domain augmentation involves creating diverse fictitious domains and simulating various clothing scenarios. Supervising features from different cloth domains enhances robustness and generalization against clothing changes. Additionally, for feature augmentation, element exchange introduces diversity concerning clothing changes. Regularizing the model with these augmented features strengthens resilience against clothing change uncertainty. Extensive experiments on cloth-changing datasets demonstrate the efficacy of our approach, consistently outperforming state-of-the-art methods. Our codes will be publicly released soon.

## CCS CONCEPTS

• **Computing methodologies** → **Object identification**.

## KEYWORDS

Domain Augmentation; Feature Augmentation; Cloth-generalized Person ReID.

## 1 INTRODUCTION

Person Re-identification (ReID) [67] is a traditional computer vision task, aiming to associate pedestrians with the same identity across different scenes. In recent years, the surge in popularity of deep learning has propelled ReID methods, leading to the development of robust models that learn discriminative features. However, many recent ReID methods have primarily addressed conventional challenges, such as adapting to human pose changes [76], handling camera view variations [39], overcoming occlusions [34, 56], and addressing Visible Infrared ReID scenarios [63, 68]. Surprisingly, these methods often overlook a crucial aspect: the possibility of pedestrians changing their clothing.

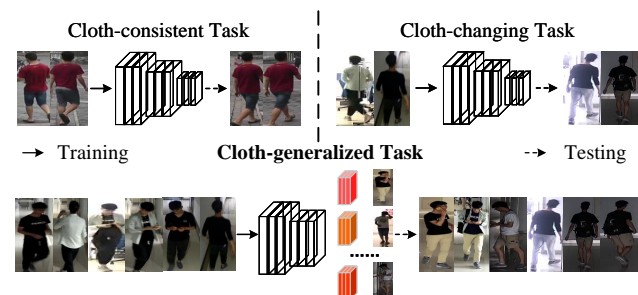

**Figure 1: Illustrating our motivation: Cloth-changing ReID tasks undergo more drastic appearance changes compared to traditional (cloth-consistent) ReID tasks. In real-world applications, both cloth-consistent and cloth-changing scenarios coexist, prompting our exploration of the cloth-generalized task. Our cloth-aware augmentation strategy, in contrast to other models, adapts features to multiple fictitious domains, enabling the learning of more generalizable and robust features.**

In real-world applications, individuals frequently change clothes in various situations, including long-term video surveillance or instances involving suspects fleeing the scene. When confronted with scenarios involving clothing changes, recent methods, primarily appearance-based, prove less effective. This limitation significantly compromises their performance, as highlighted in Figure 1. Clothing changes introduce substantial alterations in appearance, making it more challenging to re-identify individuals with the same identity. Moreover, many traditional biometric techniques become impractical in these situations, where facial features may be unclear, and gait analysis may not be feasible in surveillance videos. Hence, it is imperative to address this specific challenge [73].

In response to the cloth-changing problem, researchers have introduced datasets and explored solutions to mitigate variations caused by clothing changes. Recent cloth-changing ReID methods aim to obtain identity-discriminative and cloth-invariant feature representations, broadly categorized into three groups: (1) Designing learning strategies to mitigate the influence of clothing changes: Methods like [47] and [25] attempt to disentangle clothing-related information using various strategies. This includes adversarial-based techniques [18] and [59], as well as disentangle-based approaches [36] and [59]. (2) Leveraging auxiliary clothing-invariant modality: Methods in this category, such as [60], [47], [24], [7], [18], [8], [38], and [19], incorporate additional information like pedestrian contours, key points, and human parsing estimation results. (3) Applying augmentation strategies to enhance model robustness: Methods involve synthesizing cloth-changing images, as seen in [70] and [27], to reinforce the feature learning process.

Despite advancements, existing methods often neglect the exploration of generalization ability [82], a crucial aspect for achieving

universality in real-world applications. As depicted in Figure 1, traditional cloth-consistent ReID overlooks clothing changes that may occur when pedestrians alter their attires. Conversely, cloth-changing methods may falter in cloth-consistent cases. Moreover, when tackling ReID tasks, two critical problems arise: (1) *When only cloth-consistent data is available, how do we train a cloth-changing model? (2) How do we ensure the cloth-changing model performs well on cloth-consistent cases?* Therefore, as illustrated in Figure 1, we embark on an exploration of the cloth-generalized ReID task. This task involves training a model with the versatility to adeptly handle both cloth-consistent and cloth-changing scenarios. Recognizing the potential of augmentation strategies in improving generalization ability, we introduce Cloth-aware Augmentation (CaAug) strategies specifically tailored for both types of scenarios. Traditional augmentation methods such as cutout [15] and erasing [80] often fall short in addressing domain differences. In contrast, our approach deliberately mimics various domain gaps, allowing learned features to adapt seamlessly to diverse cloth-changing scenarios. To tackle the uncertainties associated with generative networks [61], we enhance generalization both from the latent domain and the feature aspect overhead. CaAug comprises two essential components, each contributing significantly to the model's generalization ability: Domain Augmentation (DomainAug) and Feature Augmentation (FeatAug).

Domain Augmentation (DomainAug): This component involves the generation of fictitious domains, mirroring clothing domains as depicted in Figure 1. Features learned within these domains are supervised to be both discriminative and akin to real clothing domains. It's noteworthy that we emulate actual clothing domains using four commonly employed data augmentation strategies: mixup [72], cutmix [71], cutout [15], and color jittering [52]. This deliberate choice stems from the rationale that each of these augmentation techniques introduces distinct variations in the data, simulating the diverse conditions encountered in real-world clothing domains. The advantage of incorporating these diverse augmentation strategies lies in the model's exposure to a wide range of synthetic clothing variations. To bridge the gap between fictitious and actual clothing domains, we employ domain alignment. By mimicking diverse clothing domains through these fictitious domains, our model learns discriminative clues and progressively adapts features to different outfits. This adaptability enhances the model's robustness against clothing changes by capturing shared identity-discriminative clues.

Feature Augmentation (FeatAug): We randomly interchange feature elements of the same sample (Inner-sample Augmentation) and the same identity (Inter-sample Augmentation), a strategy aimed at enriching feature diversity and effectively addressing variability in clothing changes. This intentional manipulation is grounded in the rationale that different elements within a feature set inherently carry distinct clues related to the identity being represented. By encouraging the random interchange of these elements, our approach stimulates the exploration of various facets of the feature space. The advantage of this strategy lies in its ability to foster a more comprehensive understanding of the underlying patterns and intrinsic complexities present in the data. As the model engages in the exploration of different feature elements, it becomes more adept at discerning subtle nuances and capturing the diverse manifestations of identity within the context of clothing changes. This results in a

more robust and adaptable model, ultimately enhancing its performance in cloth-changing re-identification scenarios by capturing a broader spectrum of identity-related information.

Hence, the benefits of our CaAug approach are as follows: (1) Versatility Across Tasks: This strategy can be readily applied to various tasks, including scenarios where clothing consistency is crucial. It also enhances generalization capabilities across different conditions. (2) Comprehensive Approach: In contrast to simple data augmentation, our method is more comprehensive, as it directly transforms features, enhancing both robustness and generalizability. In summary, our contributions encompass four key aspects:

We initiate the first exploration of CG-ReID and introduce a Cloth-aware Augmentation (CaAug) strategy to tackle this task. The core idea is to enhance the generalization ability of learned features without the need for additional information.

We create fictitious domains to simulate clothing domains, enabling our model to learn features that are generalizable across various cloth-changing scenarios and more comprehensive in nature.

We extend our approach by incorporating feature augmentation, which involves random interpolation of different samples. This enhances feature diversity and equips learned features to handle the uncertainties associated with varying degrees of clothing changes, further enhancing generalization ability.

We conduct extensive evaluations of our method across cloth-changing, cloth-consistent, and cloth-generalized scenarios. Our method consistently outperforms state-of-the-art techniques, as demonstrated by a robust set of experimental results.

## 2 RELATED WORKS

### 2.1 Person Re-identification

Person re-identification [67] has attracted much attention in real-world surveillance systems, which aim to associate the same pedestrian under various scenarios in different camera environments. Note that this task only relies on flexible surveillance videos, and it is more feasible for human-relevant applications compared with other biometric techniques. For example, in many scenarios, this additional bio-information is usually unavailable in real-world applications.

**Cloth-consistent Person Re-identification.** Person Re-identification [68, 86] has been explored much in recent years, which aims to learn robust features against multiple variations. According to the addressed challenges, traditional ReID methods can be roughly listed: 1) Methods for camera view or pose changes [40, 45, 58, 74, 76]. These methods aim to learn robust features against view/pose changes, which usually take vantage of domain adaptation methods. For example, VCFL [40] takes advantage of adversarial learning for learning view-invariant identity-discriminative features. 2) Methods for occlusion [16, 23, 35, 44, 54]. Methods in this category aim to solve the challenging task as the appearance varies substantially with various obstacles, especially in the crowd scenario. For example, PVPM [16] aims to learn discriminative features with pose-guided attention and part-aware visibility. 3) Methods for modality changes [5, 11, 30, 37, 63–66]. These methods aim to match the same pedestrian between the visible and infrared cameras and solve modality gaps. With the development of deep learning, more and more challenges have been widely explored. However, these methods are vulnerable to clothes changes more or less, because they rely

heavily on appearance and do not explicitly consider the clothing changes. Therefore, researchers begin to explore the cloth-changing person re-identification and aim to learn robust features against cloth changes, which is important in long-term surveillance.

**Cloth-changing Person Re-identification.** In recent years, there have been several cloth-changing datasets for paving ways: LTCC [47], PRCC [60], Celeb-ReID [25], COCAS [69], VC-Clothes [53] and CCVID [18]. Recent methods aim to learn features robust against cloth changes, which can be roughly categorized into three groups: 1) Removing cloth-variant information: For example, CESD [47] removes clothing-related information and focuses on body type information that is unrelated to view/pose. CASE [36] learns body-structural visual representations via adversarial learning and structural disentanglement. AFDNet [59] disentangle identity-related and identity-unrelated features with intra-class reconstruction and inter-class adversary. 2) Benefiting from cloth-variant information: For example, SPT [60] uses contour sketches to learn discriminative features because the shape of pedestrians won't change much with cloth-changing. 3DSL [7] leverage SMPL [42] for learning texture-insensitive 3D features. GI-ReID [28] learns cloth-agnostic representations by leveraging personal unique and cloth-independent gait information. 3) Benefiting from augmentation: For example, PosNeg [27] designs both positive and negative augmentations for enriching diversity. CCFA [20] augments feature with different semantic directions in the feature space.

Current approaches primarily focus on cloth-changing tasks, overlooking the potential for mining generalizable information. Besides, augmentation-based methods, while effective, may introduce varying degrees of noise during the augmentation process, limiting their comprehensiveness. In response, CaAug addresses these limitations: (1) Capturing Generalizable Clues: Unlike previous methods, CaAug aims to capture shared generalizable clues among multiple fictitious domains, enhancing the model's ability to generalize across diverse conditions. (2) Enhancing Features with Generalizable Information: Instead of introducing meaningless information, CaAug enhances features by introducing exclusive information within the same sample and identity through random transformations at the feature level.

## 2.2 Augmentation-based Methods

In the ReID field, data augmentation strategies have been widely explored. GAN-based methods [13, 57, 81] and VAE-based methods [58] aim to generate multiple samples. Cutout [15], CutMix [71] and MixUp [72] are all beneficial for increasing the diversity of samples. However, some data augmentation may distort the discriminative clues in pedestrian images as mentioned in [61]. Besides, generative networks usually need large computational costs. We originally proposed domain augmentation, which alleviates the potential risk that the quality of augmented data ruins learning discriminative features. In [32], the effectiveness of feature augmentation is validated. We further design a feature augmentation strategy tailored for cloth-generalized tasks, which improves the diversity of features and handles the uncertainty that the cloth-changing degree varies.

## 2.3 Variational Auto-Encoder

Variational Auto-Encoder (VAE) [29] is an important generative neural network. It consists of an encoder that aims to reduce dimensions and a decoder that aims to decrease the differences between original and output contents. Besides, VAE is capable of disentangling the latent factors of variations from the abstract representations. There are many methods of improving VAE for better handling disentangling problems. Abdi [1] focuses on representation learning and disentanglement through developing a VAE library that consists of a kind of improved VAE model. Seitzer [49] uses VAE to disentangle latent factors of variation. This paper pioneers the use of Variational Autoencoders (VAE) for generating fictitious domains: (1) Unlike Generative Adversarial Networks (GAN) and diffusion models, VAE offers a straightforward and cost-effective approach to generating domains with distinct distributions. (2) In the context of cloth-generalized problems, VAE facilitates the disentanglement of useful information for fictitious domains, improving the model's ability to capture relevant features for domain generalization.

## 3 OUR APPROACH

Under the cloth-generalized setting, pedestrian appearances may change greatly, and therefore the appearance-based ReID methods become unreliable. Our goal is to learn robust features that resist clothes changes. To achieve this, we designed the Cloth-aware Augmentation (CaAug) strategy to alleviate the variations caused by clothes changes. As illustrated in Figure 2, the whole framework includes Domain Augmentation (DomainAug) for latent domain learning and Feature Augmentation (FeatAug) for increasing the diversity of features. We first introduce the methodology details of the proposed strategy and then demonstrate the formulation of the whole model employed in this method.

### 3.1 Domain Augmentation

Recent methods for changing clothes rely on either clothing-specific information or other information unrelated to clothing, which is often inaccessible in real-world applications. We propose the domain augmentation strategy as an alternative approach, aimed at enhancing features without the need for additional information. The core concept of domain augmentation is to adapt features to various conditions, thereby promoting their generalizability and resilience to significant variations induced by changes in clothing. Moreover, domain augmentation is theoretically grounded. It is posited that data augmentation can be approximated through first-order feature averaging and second-order variance regularization components [12]. Consequently, regularizing the distribution of latent spaces with augmented data provides a more appropriate means to capture essential cues while mitigating the introduction of excessive noise. Specifically, this strategy comprises three key steps: Domain Generation, Domain Alignment, and Domain Feature Learning.

**Domain Generation**: For domain generation, we generate fictitious domains by transforming the original domain, so that the fictitious can maintain some discriminative clues. To achieve this, we simply use variational auto-encoder (VAE) [29], mainly for two reasons: 1) VAE can generate multiple domains with different distributions, which meets our demands. 2) VAE is capable of disentangling the latent factors of variations from the abstract representations. The VAE network consists of an encoder network *Enc* and a decoder network *Dec*. Given $N$ samples $X = \{x_i\}_{i=1}^{N}$, we can obtain the feature representations $F = \{f_i\}_{i=1}^{N}$ and logits $G = \{g_i\}_{i=1}^{N}$ with the

**Figure 2: Illustration of our Cloth-aware Augmentation (CaAug) strategy. It comprises two key components:** *Domain Augmentation* **consists of three steps: 1) Domain Generation: generating** $D$ **fictitious domains with** $D$ **VAEs. 2) Domain Alignment: supervising the fictitious domains to be similar to different clothes domains. 3) Domain Feature Learning: supervising the learning of features in different fictitious domains, aiming to learn robust and generalizable features.** *Feature Augmentation* **aims at enhancing feature diversity and managing uncertainties associated with cloth-generalized problems. This is achieved through the incorporation of inner-sample and inter-sample random interpolation techniques.**

backbone network. Then, we can obtain the latent codes $L = \{l_i\}_{i=1}^{N}$ through the encoder network $Enc$ and the reconstructive features $R = \{r_i\}_{i=1}^{N}$ through the decoder network $Dec$. The whole process can be formulated as below:

$$l_i = Enc(f_i), r_i = Dec(l_i). \tag{1}$$

Then, we can obtain $D$ fictitious domains using $D$ VAEs. In the $d-th$ domain, the latent features $L^d = \{l_i^d\}_{i=1}^{N}$ and the reconstructive features $R = \{r_i^d\}_{i=1}^{N}$ are included.

**Domain Alignment**: To achieve the goal that the fictitious domains can simulate different clothes domains, three demands should be satisfied: 1) Clothes domains should be related to clothes appearance, which means features in different clothes domains can be transformed back. 2) Different clothes domains should have different distributions from each other. 3) Features in different clothes domains should share some common information such as body shape, and body contour. Therefore, we use the generated images with different clothes to guide and align the fictitious distributions. First, the intuitive change of cloth-changing is color-changing, therefore we generate the images with different colors as guidance [10, 84].

Note that color affects the appearance more compared with the effect of accessories since color occupies the main part of the pedestrian's appearance. Therefore, for each sample $x_i$, we also generate guiding samples with different augmentation strategies including color-jittering [52]. The feasibility of generating images with different colors has been proved in [2], and it is a straightforward manner. Second, we can obtain the corresponding features $F^d = \{f_i^d\}_{d=1}^{D}$ and logits $G^d = \{g_i^d\}_{d=1}^{D}$ by passing the generated samples through the backbone network. Third, we use the features of the generated samples to guide the learning of fictitious domains, aiming to narrow the gaps. We use KL divergence for domain alignment because of its ability to measure the distribution gaps.

$$\mathcal{L}_{align} = \frac{1}{D} \sum_{d=1}^{D} KL(F^d, R^d), \tag{2}$$

where $KL(\cdot)$ denotes the KL divergence operation. Compared with using the generated samples for data augmentation, our domain augmentation has three advantages: 1) The features of the generated samples ($F^d$) do not back-propagate gradients since they are used as a reference distribution, and therefore we can save much

computational cost. 2) We consider the overall distribution of the generated samples rather than their discriminative clues, therefore the quality of them does not influence the overall performance much. 3) Different kinds of augmentation methods are aggregated in this manner, benefiting from the capturing of shared generalizable clues.

**Domain Feature Learning**: Our goal is to obtain robust features that preserve discrimination under multiple conditions. To supervise the features in fictitious domains, we introduce the commonly used triplet loss. First, we give the formulation of triplet loss:

$$\ell_{tri}(f_i, f_j, f_k) = [h(f_i, f_j) - h(f_i, f_k) + m]_+, \quad (3)$$

where $f_i, f_j, f_k$ form the triplets. $h(\cdot)$ is the commonly used Euclidean distance, and $m$ is the margin for triplets. Then, the triplet loss for the fictitious domains can be formulated:

$$\mathcal{L}_{tri} = \sum_{d=1}^{D} \sum_i \ell_{tri}(f_i^d, f_j^d, f_k^d). \quad (4)$$

The formulation of domain augmentation is described:

$$\mathcal{L}_{D-Aug} = \mathcal{L}_{align} + \mathcal{L}_{tri}. \quad (5)$$

## 3.2 Feature Augmentation

It has been proved that feature augmentation can improve the generalization ability [32]. To enhance the robustness of features against clothes changes, we further design the Feature Augmentation strategy, which mainly handles the uncertainty in the cloth-generalized problem. The uncertainty in the cloth-generalized problem mainly lies in two terms. 1) Under a cloth-unchanging setting, different pedestrians may look similar to the same pedestrian. This problem is more severe in a cloth-generalized setting, because the same pedestrian may wear different clothes while different pedestrians are with similar clothes. 2) The degree of clothes changes varies a lot, for example, some people only change their shirts while some people change the whole suit. To cope with this problem, we design the FeatAug with random interpolation.

**Random Interpolation.** This process is tailored for the cloth-generalized task. The synthesized features are relevant to clothes changes, thus the random interpolation is suitable for handling the uncertainty in the cloth-generalized problem. This process enlarges the feature diversity using the linear combination of two selected instance features with different suits. In the first step, two feature selection manners are included for better harnessing identity-relevant clues. 1) *Inner-sample*: we select the features (F) and randomly chosen refactoring codes ($R$) of the same sample. 2) *Inter-sample*: we select the features of the same pedestrian into different groups from each sampled training batch. Then, we synthesize two augmented features sets $\{f_i'\}_{i=1}^N$ and $\{f_i''\}_{i=1}^N$. Each sample feature $f_i'$ is generated by the interpolation of selected instance pair $< f_i, f_j >$ with length $t$ while $f_i''$ is generated by the interpolation of selected instance pair $< f_i, r_i >$, we random set $m * t$ ($m \in [0, 1]$) bits as ones and the others as zeros to form the indicator vector $I$. Then the random interpolation for feature elements is formulated:

$$f_i' = L_2(I \cdot f_i + (\mathbf{1} - I) \cdot f_j), f_i'' = L_2(I \cdot f_i + (\mathbf{1} - I) \cdot r_i), \quad (6)$$

where $\mathbf{1}$ is the all one vector and $L_2(\cdot)$ is the normalization function. We use triplet loss to supervise the augmented features:

$$\mathcal{L}_{F-Aug} = \ell_{tri}(f_i', f_j', f_k') + \ell_{tri}(f_i'', f_j'', f_k''). \quad (7)$$

## 3.3 Loss Functions

For the backbone network, we use the commonly used ResNet-50 network and choose CAL [18] and AGW [67] as the baseline. The baseline supervision losses $\mathcal{L}_B$ are triplet loss $\ell_{tri}$ and classification loss $\ell_{cls}$ as in CAL [18]. Domain Augmentation (Domain-Aug) is supervised by $\mathcal{L}_{D-Aug}$, aiming to learn robust features that are generalizable to multiple clothes domains. For Feature Augmentation, we use triplet loss $\mathcal{L}_{F-Aug}$ to supervise these augmented features, aiming to increase the diversity of features and handle the uncertainty of cloth-generalized problems.

$$\mathcal{L} = \mathcal{L}_B + \lambda_1 \mathcal{L}_{D-Aug} + \lambda_2 \mathcal{L}_{F-Aug}. \quad (8)$$

## 3.4 Implementation Details

**Network Configuration.** Throughout all experiments, we utilize CAL [18] and AGW [18] for different cloth-generalized cases. This choice underscores the versatility of our method as an easily integrable module for enhancing generalization capabilities. For domain generation (with $D = 4$), we employ four commonly used data augmentation strategies and utilize four VAE networks to generate corresponding fictitious domains. The lengths of latent codes are set equal to the number of classes, ensuring they adequately represent the distribution. Feature augmentation comprises Random Interpolation, with $m$ set to 0.1 to regulate the number of exchanged bits. These parameters are carefully selected through parameter analysis (refer to the appendix for further details).

**Training.** The learning rate and the training strategy are the same as that of CAL [18]. For Domain Alignment: we use the backbone to extract features with the frozen model since they are used as the reference distributions, and KL divergence is used for alignment. For Domain Feature Learning: we supervise the reconstructed features with triplet loss in fictitious domains, and the margin is set as 0.3. For feature augmentation, the augmented features are obtained through synthesizing features in the original domain and supervised by triplet loss with a margin set as 0.3.

**Inference.** In the inference phase, neither domain augmentation nor feature augmentation is necessary. Consequently, the trained model is employed solely to extract features from query and gallery images, facilitating feature matching.

## 4 EXPERIMENTS

## 4.1 Datasets and Evaluation Protocol

To cope with the cloth-changing and cloth-generalized problem in the ReID field, two datasets are used for performance evaluation, including LTCC [47] and PRCC [60]. A cloth-consistent dataset (Market1501 [77] is used to prove the effectiveness of our method for cloth-unchanged tasks.

**LTCC** [47] contains 17,138 person images of 152 identities, which can be divided into two subsets: one cloth-changing set where 91 persons appear with 417 different sets of outfits in 14,756 images, and one cloth-consistent subset containing the remaining 61 identities with 2,382 images without outfit changes.

**PRCC** [60] consists of 221 identities with three camera views. Each person in Cameras A and B wears the same clothes, and the

**Table 1: Comparison of Rank-k and mAP Performance with State-Of-The-Art (SOTA) Methods in LTCC and PRCC.**

| Method | Venue | LTCC | | | | PRCC | | | |
|---|---|---|---|---|---|---|---|---|---|
| | | General | | Cloth-changing | | Same-clothes | | Cross-clothes | |
| | | Rank1 | mAP | Rank1 | mAP | Rank1 | mAP | Rank1 | mAP |
| HACNN [33] | CVPR18 | 60.2 | 26.7 | 21.6 | 9.3 | 82.5 | 84.8 | 21.8 | 23.2 |
| PCB [51] | ECCV18 | 65.1 | 30.6 | 23.5 | 10.0 | 99.8 | 97.0 | 41.8 | 38.7 |
| MGN [55] | ACMMM18 | 68.4 | 34.6 | 25.0 | 12.6 | 98.2 | 98.4 | 53.5 | 53.3 |
| ISP [85] | ECCV20 | 66.3 | 29.6 | 27.8 | 11.9 | 92.8 | - | 36.6 | - |
| AGW [67] | TPAMI21 | 61.4 | 21.5 | 27.0 | 8.4 | 97.8 | 91.5 | 44.7 | 37.1 |
| SPT+ASE [60] | TPAMI19 | - | - | - | - | 64.2 | - | 34.4 | - |
| CESD [47] | ACCV20 | 71.4 | 34.3 | 26.2 | 12.4 | - | - | - | - |
| 3DSL [7] | CVPR21 | - | - | 31.2 | 14.8 | - | - | 51.3 | - |
| FSAM [24] | CVPR21 | 73.2 | 35.4 | 38.5 | 16.2 | - | - | - | - |
| GI-ReID [28] | CVPR22 | 63.2 | 29.4 | 23.7 | 10.4 | 80.0 | - | 33.3 | - |
| CAL [18] | CVPR22 | 74.2 | 40.8 | 40.1 | 18.0 | 100.0 | 99.8 | 55.2 | 55.8 |
| AIM [62] | CVPR23 | 76.3 | 41.1 | 40.6 | 19.1 | 100.0 | 99.9 | 57.9 | 58.3 |
| CCFA [20] | CVPR23 | 75.8 | 42.5 | 45.3 | 22.1 | 99.6 | 98.7 | 61.2 | 58.4 |
| SCNet [19] | ACMMM23 | 76.3 | 43.6 | 47.5 | 25.5 | 100.0 | 97.8 | 61.3 | 59.9 |
| CaAug (AGW) | | 75.4 | 37.1 | 38.8 | 17.0 | **100.0** | **99.9** | 55.5 | 56.4 |
| CaAug (CAL) | | **78.3** | **45.9** | **50.5** | **25.8** | 99.9 | 98.7 | **63.9** | **60.1** |

person wears different clothes in Camera C. In general, approximately 152 images of each person are included in this dataset, for a total of 33698 images.

**Market1501** [77] contains 32,668 bounding boxes with six cameras. The training set includes 12,936 images of 751 identities while the testing set includes 19,732 images of 750 identities.

**DukeMTMC-reID** [48] is a subset of the large-scale multi-target pedestrian tracking dataset DukeMTMC for image-based ReID. It contains 16,522 images of 702 identities for training and 17,661 images of 702 identities for testing.

**Evaluation Protocol** follows the same protocol as the original protocol of the dataset. For LTCC, to better analyze the results of long-term cloth-changing Re-ID in detail, we introduce two test settings: standard-setting (we use all the data for training) and cloth-changing setting (we only match the cloth-changing pedestrians ) as in [47]. For PRCC, we also introduce two test settings: cross-clothes (images of camera A and camera C are used as the gallery and query sets) and same-clothes (images of camera A and camera B are used as the gallery and query sets). Cumulative Matching Curves (CMC) [17] and mean Average Precision (mAP) [3] are used as the evaluation metrics.

## 4.2  Ablation Study

To gain more insights into CaAug, we conduct a set of ablative studies on LTCC [47] and PRCC [60], with ResNet-50 as the backbone network. Specifically, we analyze the influence of Domain Augmentation (DomainAug) and Feature Augmentation (FeatAug). When we combine the two components, we obtain the best performance. This suggests that these modules are complementary to each other, and confirms the effectiveness of our whole design.

**Effectiveness of Domain Augmentation (DomainAug)**. First, we examine the impact of domain augmentation using $\mathcal{L}_{D-Aug}$ (Eq.(5)). The results are summarized in Table 2. For LTCC, we observe a significant improvement of 2.7% and 4.3% in Rank1 through

**Table 2: Comparison Rank-k and mAP Performance with State-Of-The-Art (SOTA) methods in Market1501 (Market).**

| Method | Venue | Market | |
|---|---|---|---|
| | | Rank1 | mAP |
| MLFN [4] | CVPR18 | 90.0 | 74.3 |
| HACNN[33] | CVPR18 | 91.2 | 75.7 |
| PCB [51] | ECCV18 | 92.3 | 77.4 |
| Part-aligned [50] | ECCV18 | 93.8 | 79.9 |
| MGN [55] | ACMMM18 | 95.7 | 86.9 |
| DG-Net [79] | CVPR19 | 94.4 | 85.2 |
| BOT [43] | CVPR19 | 94.5 | 85.9 |
| ISP [85] | ECCV20 | 95.3 | 88.6 |
| SCSN [9] | CVPR20 | 95.7 | 88.5 |
| SBS [22] | arxiv20 | 95.7 | **89.3** |
| CDNet [31] | CVPR21 | 95.1 | 86.0 |
| AGW [67] | TPAMI21 | 95.1 | 87.7 |
| PGFL-KD [75] | ACMMM21 | 95.3 | 87.2 |
| Celeb [25] | IJCNN19 | 91.2 | 77.2 |
| ReIDcaps [26] | TCSVT19 | 92.8 | 78.0 |
| CASE-Net [36] | WACV20 | 94.6 | 85.7 |
| FSAM [24] | CVPR21 | 94.6 | 85.6 |
| CAL [18] | CVPR22 | 94.7 | 87.5 |
| CaAug (CAL) | | 95.1 | 88.3 |
| CaAug (AGW) | | **95.9** | 89.2 |

a comparison between the baseline (CAL [18]) with and without DomainAug. This suggests that learned features become more robust and comprehensive by adapting to multiple fictitious domains. To further validate the superiority of the proposed DomainAug, we compare it with directly applying data augmentation, as shown in Table 4. The results indicate that our strategy effectively mitigates the potential issue of augmenting data with poor quality. Moreover,

**Table 3: Ablation studies on different components of our method in LTCC with the CAL Baseline.**

| $\mathcal{L}_{D-Aug}$ | $\mathcal{L}_{F-Aug}$ | General | | Cloth-changing | |
|---|---|---|---|---|---|
| | | Rank1 | mAP | Rank1 | mAP |
| - | - | 74.2 | 40.8 | 40.1 | 18.0 |
| ✓ | - | 76.9 | 44.6 | 44.4 | 23.2 |
| ✓ | ✓ | 78.3 | 45.9 | 50.5 | 25.8 |

**Table 4: The Comparison with Other Augmentation Strategies in LTCC with the CAL Baseline.**

| Strategy | General | | Cloth-changing | |
|---|---|---|---|---|
| | Rank1 | mAP | Rank1 | mAP |
| Domain Augmentation | 76.9 | 44.6 | 44.4 | 23.2 |
| Mixed Augmentation | **78.3** | **45.9** | **50.5** | **25.8** |
| CutOut [15] | 74.6 | 40.2 | 37.5 | 17.1 |
| CutMix [71] | 73.4 | 39.6 | 35.7 | 16.7 |
| Mixup [72] | 74.4 | 40.9 | 39.0 | 18.5 |
| Color Jittering [52] | 75.1 | 41.5 | 40.6 | 19.0 |

our domain augmentation approach saves computational resources by utilizing a frozen model and serves as an efficient integration strategy for various augmentation techniques.

**Effectiveness of Feature Augmentation (FeatAug).** Second, we investigate the feature augmentation with $\mathcal{L}_{F-Aug}$ (Eq.(7)). Concretely, for the LTCC dataset, we compare the results of the baseline CAL [66] with/without $\mathcal{L}_{F-Aug}$ to evaluate its effectiveness. As shown in Table 2, we gain 1.4% and 6.1% on Rank1 separately. This indicates that we can improve the diversity of features and handle the uncertainty of the cloth-changing problem, demonstrating that our feature augmentation strategy is suitable for cloth-generalized tasks. Besides, we also conduct experiments to explore the performance of random noise and random interpolation.

**Parameter Analysis.** In our experiments, we explore the effect of the loss weight parameter $\lambda_1$ and $\lambda_2$ by systematically varying its values from 0 to 1 in increments of 0.1. The results, as shown in Figure 3(a) and Figure 3(b), clearly demonstrate consistent improvements over the baselines. Therefore, we set $\lambda_1 = 1$ and $\lambda_2 = 1$. This indicates that finding appropriate values for loss weights plays a crucial role in optimizing the overall performance of our method in cloth-generalized ReID tasks.

## 4.3 Comparison with State-Of-The-Art methods

We compare our method with cloth-changing methods and cloth-consistent methods over multiple cases (**cloth-changing cases**, **cloth-consistent cases**, and **cloth-generalized cases**), and prove its superiority over discrimination and generalization ability.

**1. Cloth-Changing Cases.** These cases focus on the robustness against cloth changes, and we conduct experiments on two commonly used cloth-changing datasets.

**LTCC** [47]. We evaluate our proposed method on the LTCC dataset and compare it with several competitors. In Table 1, we report the Rank1 and mAP, from which several observations can be made: (1) Traditional ReID methods (designed for cloth-consistent

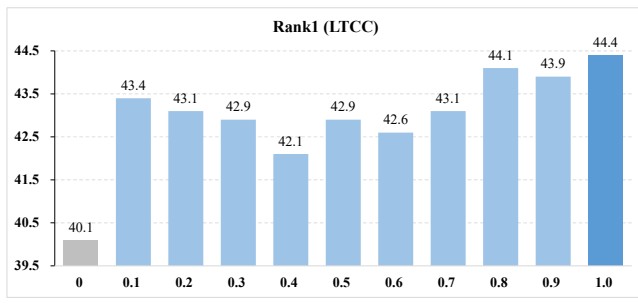

(a) Loss Weight for Domain Augmentation

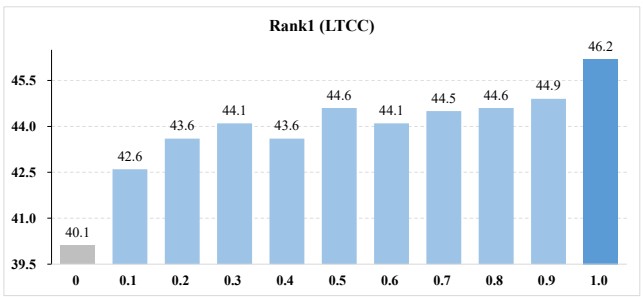

(b) Loss Weight for Feature Augmentation

**Figure 3: Parameter Analysis of Feature Augmentation in LTCC. We find that the feature augmentation strategy consistently improves performance. Through (a) and (b), we find that setting hyper-parameters $\lambda_1 = 1$ and $\lambda_2 = 1$ is better.**

scenarios) such as AGW [67] are significantly impacted by the cloth-changing problem. Our method notably improves upon AGW, indicating its effectiveness in mitigating the negative impact of clothing changes and enhancing feature robustness. (2) By following the same protocol of LTCC, we report results for both general and cloth-changing settings, demonstrating the superiority of our method. For instance, compared to the recent cloth-changing method SCNet [19], we achieve an additional 2% gain in Rank1.

**PRCC** [60]. Following the protocol of PRCC, we report the Rank1 and mAP results under cross-clothes and same-clothes settings, which are shown and compared in Table 1. We have the following observations: (1) AGW [67] fails to achieve competitive performance over the cloth-changing setting. Our method achieves a significant increase with both Rank1 and mAP evaluation. (2) Compared with other cloth-changing methods, our method reaches SOTA under mAP and Rank1 evaluation metrics, which suggests the effectiveness of our method.

**2. Cloth-Consistent Cases.** These cases focus on the robustness against traditional ReID challenges (such as illumination, view changes, and so on). These cases are also crucial since they are also general in real-world applications, and we conduct experiments on a commonly used cloth-consistent dataset.

**Market1501**[77]. To validate that our method is also suitable for cloth-consistent cases, we further conduct experiments on Market1501 (Market). As shown in Table 2, we can conclude that our

**Table 5: The Comparison of Rank1 and mAP Performance with Other Methods for The Generalization Ability within Cloth-Consistent Datasets.**

| Method | Venue | Market → Duke | |
|---|---|---|---|
| | | Rank1 | mAP |
| ResNet50 [21] | CVPR16 | 29.1 | 15.6 |
| IDE [78] | CVPR17 | 38.4 | 22.0 |
| SPGAN [14] | CVPR18 | 41.1 | 22.3 |
| PCB [51] | ECCV18 | 43.3 | 25.2 |
| MGN [55] | ACMMM18 | 56.6 | 37.4 |
| BOT[43] | CVPRw19 | 43.9 | 26.1 |
| SBS [22] | arxiv20 | 54.1 | 32.9 |
| APNet [6] | TIP21 | 37.7 | 22.8 |
| FA-Net [41] | TIP21 | 49.3 | 30.7 |
| AGW [67] | TPAMI21 | 53.4 | 33.4 |
| Pos-Neg [27] | TIP22 | 55.8 | 36.4 |
| CAL [18] | CVPR22 | 8.3 | 3.4 |
| CaAug (AGW) | | **57.2** | **36.5** |
| CaAug (CAL) | | 36.6 | 20.5 |

**Table 6: The Comparison of Rank1 and mAP Performance with Other Methods for The Generalization Ability cross Cloth-Changing and Cloth-Consistent Datasets.**

| Method | Venue | LTCC → Market | |
|---|---|---|---|
| | | Rank1 | mAP |
| ResNet50 [21] | CVPR16 | 24.7 | 9.6 |
| HACNN [33] | CVPR18 | 26.9 | 10.4 |
| PCB [51] | ECCV18 | 31.2 | 13.4 |
| MGN [55] | ACMMM18 | 47.3 | 21.6 |
| OSNet [83] | ICCV19 | 34.3 | 15.6 |
| BOT [43] | CVPR19 | 42.4 | 19.5 |
| MuDeep [46] | TPAMI20 | 29.4 | 11.2 |
| SBS [22] | arxiv20 | 34.9 | 14.7 |
| CESD [47] | ACCV20 | 37.4 | 17.0 |
| AGW [67] | TPAMI21 | 46.7 | 20.7 |
| Pos-Neg [27] | TIP22 | 48.2 | 22.6 |
| CAL [18] | CVPR22 | 38.5 | 18.8 |
| CaAug (AGW) | | **48.8** | **23.8** |
| CaAug (CAL) | | 42.8 | 19.4 |

**Table 7: The Comparison of Rank1 and mAP Performance with Other Methods for The Generalization Ability.**

| Method | Venue | LTCC → PRCC | |
|---|---|---|---|
| | | Rank1 | mAP |
| MGN [55] | ACMMM18 | 29.5 | 40.5 |
| BOT [43] | CVPR19 | 30.1 | 39.6 |
| SBS [22] | arxiv20 | 28.9 | 39.3 |
| AGW [67] | TPAMI21 | 31.5 | 42.0 |
| Pos-Neg [27] | TIP22 | 31.6 | 42.5 |
| CAL [18] | CVPR22 | 37.9 | 36.3 |
| CaAug (AGW) | | 33.5 | **42.8** |
| CaAug (CAL) | | **42.3** | 38.1 |

method is also suitable for cloth-consistent cases through the comparison with cloth-unchanging and cloth-changing methods. The rationale lies in two terms: 1) Domain Augmentation can still improve the generalization ability and robustness of features under cloth-consistent conditions, for adapting features to multiple fictitious domains. 2) Feature Augmentation can still increase the diversity of features with the introduction of random noises and capture more discriminative information with random interpolation.

**3. Cloth-Generalized Cases**. These cases focus on the generalization ability, which is relevant to the scope of the application. We design three settings that can roughly represent the generalizable setting over cloth-changing and cloth-consistent datasets: Market1501 → Duke, LTCC → Market1501, LTCC → PRCC.

**Market1501** [77] → **DukeMTMC-reID** [48]. This setting addresses scenarios involving both cloth-consistent and cloth-changing cases. In Table 5, we provide a comparison of our method with other cloth-consistent techniques, demonstrating clear improvements. Additionally, we compare the results of **Market1501** [77] → **LTCC** [47] with CAL and our method, yielding 15.6% and 5.1% versus 35.7% and 12.3% in rank1 and mAP, respectively. These findings underscore the enhanced generalization capability of our approach, particularly highlighting its efficacy in scenarios requiring the learning of a cloth-generalized model with only cloth-consistent data available.

**LTCC** [47] → **Market1501**[77]. This setting effectively encompasses scenarios spanning both cloth-consistent and cloth-changing cases. Table 6 demonstrates the efficacy of our method, showcasing improvements. These results indicate our ability to discern discriminative cues solely from cloth-changing data, thereby benefiting scenarios where ensuring performance in the presence of unknown cloth-changing occurrences is crucial.

**LTCC** [47] → **PRCC**[60]. This setting captures scenarios involving cloth-changing cases. Table 7 presents qualitative comparisons within cloth-changing cases, clearly demonstrating our ability to learn generalizable features that adapt to various cloth-changing scenarios. These findings are particularly beneficial in scenarios where pedestrians exhibit significant diversity in clothing styles.

## 5 CONCLUSION

We seek to tackle the formidable cloth-generalized problem in person ReID, an area where existing cloth-changing methods often overlook the extraction of generalizable insights. In this paper, we introduce the Mixed Augmentation strategy, designed to guide feature learning and cultivate resilient features resistant to changes in clothing. Our approach involves generating multiple fictitious domains to emulate clothing variations and adapting features to these domains, thereby fostering the acquisition of robust and versatile features. Additionally, to address the uncertainty inherent in the cloth-generalized problem, we synthesize features by incorporating inner-sample and inter-sample random interpolation techniques, while also supervising the learning of augmented features. While our method is tailored for the cloth-generalized problem, we firmly believe that our model possesses the capability to contend with even more intricate conditions characterized by significant variations.

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
