# OpenReview forum: "Cloth-aware Augmentation for Cloth-generalized Person Re-identification"
_acmmm.org/ACMMM/2024/Conference — MM2024 Oral_

### Official Review · Reviewer_ptdi · 2024-05-11

**Rating:** 5
**Confidence:** 4

**Summary:**

In summary, the paper introduces a novel task of cloth-generalized person re-identification (CG-ReID) and proposes a Cloth-aware Augmentation (CaAug) strategy that comprises domain augmentation and feature augmentation to learn identity-relevant features that are adaptable to both cloth-consistent and cloth-changing scenarios. The experiments demonstrate the effectiveness of the approach, surpassing state-of-the-art methods on cloth-changing datasets.

**Strengths:**

1) The paper pioneers the cloth-generalized person re-identification (CG-ReID) task, bridging the gap between cloth-consistent and cloth-changing scenarios.
2) The Cloth-aware Augmentation (CaAug) strategy, comprising domain and feature augmentation, enables learning identity-relevant features robust to clothing changes.
3) Extensive experiments on cloth-changing datasets demonstrate the effectiveness of the proposed approach, outperforming existing methods.
4) The codes will be publicly released, ensuring reproducibility and facilitating further research.

**Limitations:**

1) The choice of augmentation methods is not well justified and may exclude more recent and effective techniques.
2) The use of a CNN baseline as the network architecture may limit performance.
3) Win-Win by Competition: Auxiliary-Free Cloth-Changing Person Re-Identification, TIP 2023, that serves as the predecessor of AIM [62] and would provide valuable insights for inclusion in the comparison table.
4) The evaluation of Tables 5-7 seems lack comparisons with state-of-the-art methods from 2023.

**Suitability:**

3

---

### Official Review · Reviewer_iXnX · 2024-05-14

**Rating:** 3
**Confidence:** 4

**Summary:**

The paper presents an augmentation-based method for Cloth-Generalized Person Re-Identification, in which domain augmentation and feature augmentation are performed. For domain augmentation, fictitious domains are generated using VAEs, and aligned using samples generated by popular augmentation methods. For feature augmentation, features are exchanged in inter-sample and inner-sample manners for augmented features.

**Strengths:**

- Ideas of augmentation have always been promising for CCRe-ID [1][2]
- For domain augmentation, the paper leveraged VAEs to obtain fictitious domains, which is a novel approach
- For feature augmentation, inter-sample and inner-sample feature exchange sounds effective
- Writing is quite clear and easy to follow

[1] Nguyen et al., CCPA: Long-term Person Re-Identification via Contrastive Clothing and Pose Augmentation
[2] Han et al., Clothing-change feature augmentation for person re-identification

**Limitations:**

- Cloth-Generalized Scenario is vague and not novel. Intuitively, if a model can perform well under cloth-changing scenario, it is likely to perform well on cloth-consistent scenario which is less challenging.

- The necessity and novelty of Cloth-Generalized Re-ID task is not validated. Consider choosing a SOTA CCRe-ID method, running experiment on CCRe-ID datasets, then run on standard Re-ID datasets and show performance drop for example.

- Related work in CCRe-ID lacks a review of more recent works (There are dozens of papers in CCRe-ID published in 2023 and early 2024 besides the covered papers)
- Not comprehensive experimental results

        + Similarly, in all experiments, results are compared to a very small number of previous (especially recent) methods, which limits the comprehensiveness of the effectiveness of the proposed method. More recent methods in 2023 and 2024 (see [1-5] for examples) should be taken into comparison. Eg. Tables 2, 5, 6, 7 only take into comparison methods from 2022 backward.
        + Your method leverages original RGB image only for training, while many CCRe-ID methods ([1, 3, 5] and others) also capture auxiliary modalities like pose, silhouette, SMPL shape. Specific comparison with this line of methods should be performed to further validate the superiority of the proposed method.

- Incorrect/unclear technical details

       + On lines 400-402, you stated the rationale for your used augmentation methods is to make different domains should share common body shape or contours, however, [mixup, cutout, cutmix] will distort the body shape and contour, which is a conflict.
       + Unclear where $F^d$ is obtained from. Is it from a mixed of samples generated by all 4 augmentation methods, and for each sample in a batch, a random method out of those 4 is chosen?

- In implementation details, why use 4 VAEs? Is that because you used 4 augmentation methods or a finding after trials and errors?

- The used datasets are quite small-scale. Evaluation on very large-scale datasets with diverse distributions in data like DeepChange [6] would be more convincing. This also validates generalizability of the method.

- Insufficient evaluation

       + Since the proposed method is augmentation-based, results should be specifically compared with previous augmentation-based methods to validate superiority.
       + In Table 5, 6, 7, are those results produced on the second dataset after training on the first dataset? This should be clarified.
       + In Table 5, 6, 7, the paper lacks showing the performance on the first dataset compared to on the second dataset, which helps validate the necessity of cloth-generalized Re-ID task, and the generalizability of the proposed method (e.g. shown by a smaller gap in performance produced by the proposed method compared to previous method)
       + In Table 6, compared methods belong to CCRe-ID should be noted and specifically compared to. This helps to show the superiority of the proposed method on cloth-generalized Re-ID scenario over previous CCRe-ID methods which only dealt with clothing changes without maintaining performance on cloth-consistent scenario.
       + In Table 3 and 4, experiment with FeatAug and without DomainAug would be helpful to further validate the effectiveness of FeatAug
       + On line 727, you should refer to Table 3, not Table 2
       + Feature map visualization will be very helpful to study the effectiveness of feature augmentation.

- Critical ethical issue: Duke-MTMC has been retracted and should not be used.

[1] Liu et al. Pose-Guided Attention Learning for Cloth-Changing Person Re-Identification, In IEEE TMM, 2023

[2] Huang et al. Meta Clothing Status Calibration for Long-Term Person Re-Identification. In IEEE TIP, 2024.

[3] Nguyen et al. Contrastive viewpoint-aware shape learning for long-term person re-identification. In WACV, 2024

[4] Yang et al. Win-win by competition: Auxiliary-free cloth-changing person re-identification. In IEEE TIP, 2023.

[5] Zhang et al. Multi-biometric unified network for cloth-changing person re-identification. In IEEE TIP, 2023.

[6] Xu, Peng, and Xiatian Zhu. Deepchange: A long-term person re-identification benchmark with clothes change. In ICCV. 2023.

**Suitability:**

3

---

### Official Review · Reviewer_z3cH · 2024-05-26

**Rating:** 5
**Confidence:** 4

**Summary:**

This paper proposed a novel cloth generalized person re-identification (CG-ReID) task and tackled the proposed problem with domain augmentation and feature augmentation strategies.

**Strengths:**

1. The experiments and ablations in this work are sufficient and well analysed.
2. The proposed method performs well on existing large-scale Clothes-changing benchmarks (e.g. PRCC, LTCC), with a competitive performance on traditional ReID benchmarks (e.g. Market).
3. The proposed CG-ReID setting is pratical and useful as it is highly corresponding to real- world applications. Therefore, the ideas of 'train a cloth-changing model with cloth-consistent data' and 'ensure the cloth-changing model performswell on cloth-consistent cases' are well motivated. Moreover, the general motivation was well delivered in the introduction section and easy to follow. In a word, I believe the newly proposed pratical setting would benefit further research for the ReID community.

**Limitations:**

1. From my understanding, the proposed CG-ReID should focus on tackling the generalizable across both clothes-changing and clothes consistent/ same-clothes cases. However, in the experiment part, only the LTCC → Market (clothes-changing → same-clothes) was presented in Tab.6 . Considering that the same-clothes cases also exist (cases of one pedestrain wearing the same clothing) in clothes-chaning datasets (e.g. LTCC), this is not extremely tough case,  and such missing results (same-clothes →  clothes-changing ) would weaken the contributions. As aforementioned, it is suggested to compare existing single-source domain generalizable ReID (DG-ReID) methods with the proposed CG-ReID methods.
2. For the method part, domain alignment was well explored in DG-ReID. Using feature augmentation for CC-ReID shared similar ideas with CCFA[20] and SPS[a]. Considering their roles in a newly proposed setting, It is suggested to discuss the difference between the proposed alignment method and those employed in DG-ReID and existing augmentation-based CC-ReID methods.
a: Semantic-guided Pixel Sampling for Cloth-Changing Person Re-identification 2021

**Suitability:**

3

---

### Meta-Review · Area_Chair_3LmM · 2024-06-25

**Recommendation:** Accept (Oral)
**Confidence:** 5

**Metareview:**

All reviewers are satisfied with the submission and responses. I hence recommend its final acceptance.